

# Dissecting effects of orbital drift of polar-orbiting satellites on accuracy and trends of cloud fractional cover climate data records

Jędrzej S. Bojanowski[1] and Jan P. Musiał[1]

[1]Institute of Geodesy and Cartography, Remote Sensing Centre, Modzelewskiego 27, PL02-679 Warsaw, Poland

**Correspondence:** Jędrzej Bojanowski (jedrzej.bojanowski@igik.edu.pl)

**Abstract.** Radiometers such as the AVHRR mounted aboard a series of the NOAA and MetOp polar-orbiting satellites provide 4-decade-long global climate data records (CDRs) of cloud fractional cover. Generation of such long datasets requires combining data from consecutive satellite platforms. A varying number of satellites operating simultaneously in the morning and afternoon orbits, together with the satellite orbital drift cause the uneven sampling of the cloudiness diurnal cycle along

a course of CDR. This in turn leads to significant biases, spurious trends and inhomogeneities in the data records of climate variables featuring the distinct diurnal cycle (such as clouds). To quantify the uncertainty and magnitude of spurious trends in the AVHRR-based cloudiness CDRs, we sampled the 30-minute reference CM SAF Cloud Fractional Cover dataset derived from Meteosat First and Second Generation (COMET) at times of the NOAA and MetOp satellites overpasses. The sampled cloud fractional cover (CFC) time series were aggregated to monthly means and compared with the reference COMET

dataset covering the Meteosat disc (up to 60 degrees N/S/W/E). For individual NOAA/MetOp satellites the errors in mean monthly CFC reach ±10% (bias) and ±7% per decade (spurious trends). For the combined data record consisting of several NOAA/MetOp satellites, the CFC bias is 3% and the spurious trends are 1% per decade. This study proves that before 2002 the AVHRR-derived CFC CDRs do not comply with the GCOS temporal stability requirement of 1% CFC per decade just due to the satellite orbital drift effect. After this date the requirement is fulfilled due to the numerous NOAA/MetOp satellites

operating simultaneously. Yet, the time series starting in 2003 is shorter than 30 years that voids climatological analyses. We expect that the error estimates provided in this study will allow for a correct interpretation of the AVHRR-based CFC CDRs and ultimately will contribute to the development of a novel satellite orbital drift correction methodology widely accepted by the AVHRR-based CDRs providers.

## 1 Introduction

Cloud feedback to the global warming remains one of the biggest uncertainties in climate projections. To improve comprehension of this complex physical phenomena, a long reliable time series of cloud fraction measurements are required at a global scale. In this respect, multi-decadal ground-based visual cloud observations, that have been recently supported or replaced by the ceilometers or total sky cameras, are still widely used in climatological studies. However, they are often inhomogeneous and located in densely populated regions leaving the vast oceanic areas, polar regions, high mountains, deserts, as well as trop-

ical and Taiga forests under-sampled. Despite aforementioned issues, the surface synoptic observations (SYNOP) have been





widely exploited for evaluation of satellite-based cloud climate data records (CDRs) (Meerkötter et al., 2004; Dybbroe et al., 2005; Kotarba, 2009; Eastman and Warren, 2010; Fontana et al., 2013; Musial et al., 2014; Bojanowski and Musiał, 2018; Bojanowski et al., 2018). In the last decade, novel referential cloud properties datasets derived from the active sensors such as the radar onboard the CloudSat satellite (Stephens et al., 2002) and the LiDAR onboard the CALIPSO satellite (Winker et al.,

2009) allow for analyses of the global cloud vertical structure with great sensitivity (Karlsson and Johansson, 2013; Karlsson and Håkansson, 2017; Stengel et al., 2015). Nevertheless, they are too short for the climate change studies. Another source of long-term datasets on global cloudiness originates from the climate reanalysis models (e.g. ERA-5) that assimilate both the ground measurements and satellite products. Yet, due to great complexity of cloud climatic feedback together with coarse spatial and horizontal resolutions of model grids, the cloud formation and dissipation processes are not accurately represented

in the climate models.

In the context of global cloud cover studies, only the passive satellite radiometers recorded over 30 years of data which is the minimal period to draw meaningful conclusions on global climate change (Rossow and Schiffer, 1999; Foster and Heidinger, 2013; Karlsson and Johansson, 2013). Nevertheless, these long time series (exceeding 30 years) can only be generated by merging observations acquired by several instruments such as the Advanced Very High Resolution Radiometer (AVHRR)

mounted onboard a series of NOAA and MetOp satellites. Amongst CDRs derived from the AVHRR time series there are: International Satellite Cloud Climatology Project (ISCCP, Rossow and Schiffer, 1999; Young et al., 2018) which also combines geostationary sensors, Pathfinder Atmospheres Extended (PATMOS-x, Heidinger et al., 2014), CLoud, Albedo and RAdiation dataset (CLARA-A2, Karlsson et al., 2017) of the EUMETSAT Satellite Application Facility on Climate Monitoring (CM SAF), the Community Cloud Retrieval for Climate dataset of the Cloud_cci data set (CC4CL-AVHRR, Stengel et al., 2017,

2019) generated in the framework of the European Space Agency Climate Change Initiative (ESA CCI), and the AVHRR Local Area Coverage satellite cloud climatology over Central Europe derived by means of the Vectorized Earth Observation Retrieval (VEOR) method (Musiał and Bojanowski, 2017).

A data fusion from several instruments can cause spurious temporal trends in CDRs originating from: instrument malfunction or degradation, satellite attitude instability, biases in ancillary data (e.g. modelled surface temperature), as well as the retrieval

algorithm shortcomings. Besides these problems, the remaining data sampling issue is related to the variable local time and number of satellite acquisitions during a day (Fig. 1). The former is related to the changing number of NOAA/MetOp satellites operating concurrently in orbit. The latter is caused by the satellite orbital drift that gradually lowers the satellite orbit due to the Earth's gravity. Both effects can lead to significant biases in aggregated products (e.g., Level-3 monthly means) derived from polar-orbiting satellite measurements (Salby and Callaghan, 1997). These effects influence various variables featuring a distinct

diurnal cycle such as: outgoing longwave radiation (Salby, 1982a, b; Fowler et al., 2000), stratospheric gases concentration (Salby, 1987), brightness temperature (Leroy, 2001; Kirk-Davidoff et al., 2005), as well as cloud cover (Bergman and Salby, 1996; Wylie et al., 2005; Devasthale et al., 2012; Foster and Heidinger, 2013). Many aforementioned studies elaborated on the correction of these effects but none of them focused on a detailed quantification of errors and spurious trends that are introduced in the CFC CDRs.



Devasthale et al. (2012) proposed to remove the signal related to the orbital drift delineated by the rotated empirical orthogonal function (REOF) analysis . Although the method gave promising results, it is sensitive to a decision which REOF loadings are related to the drift. Therefore, the risk of removing the real climatic signal cannot be neglected. Foster and Heidinger (2013) derived the CFC mean diurnal cycles by fitting sinusoidal function to all observations in the AVHRR record. Further, these functions were used to model the hourly CFC from instantaneous AVHRR observations, which were in turn aggregated

to monthly means. This method was applied to the PATMOS-x CDR and resulted in the diurnally-corrected CFC monthly means featuring coarse 1×1 deg spatial resolution. Yet, none of the described methods was recommended and applied for a CFC CDRs generation within the major European frameworks providing the satellite-derived geophysical data sets suitable for climate monitoring (i.e. ESA CCI or CM SAF). Moreover, an assessment of the impact of orbital drift on CFC CDRs was listed in the mission statements of the International Cloud Working Group (ICWG) within the Coordinated Group for Meteorological

Satellites (Wu et al., 2017). In this context, the unprecedented quantification presented in this study gives a comprehensive, yet missing perspective on the magnitudes of errors and spurious trends in the AVHRR CFC CDRs introduced by the satellite orbital drift and variable sampling of a cloudiness diurnal cycle.

The sparse sampling of a diurnal cycle combined with the satellite orbital drift can lead to unreliable mean estimates and introduction of spurious temporal trends in the Level-3 CDRs. The quantification of these effects on a CDR is complex, as

they depend on: (1) geographic location, (2) amplitude and phase of the diurnal cycle of a measured variable, (3) local time when the diurnal cycle is sampled by satellite observations, (4) magnitude of the satellite orbital drift, (5) number of available satellite observations per day, and (6) measurements selection and aggregation methodologies applied to derive the Level-3 product. In this respect, the aim of the presented study is to quantify the impact of the AVHRR satellite orbital drift combined with variable sampling of a cloudiness diurnal cycle on the accuracy of the cloudiness CDRs and to quantify the magnitude of

spurious temporal trends. To reach this aim, the approach applied builds on a referential CDR with fully resolved cloudiness diurnal cycle with 30-minutes sampling derived from the geostationary satellite measurements covering the Meteosat disc.

## 2   Data

The analyses were based on two cloud fractional cover CDRs derived from polar-orbiting (CLARA-A2) and geostationary satellites (COMET) featuring coarse and dense temporal sampling of cloudiness diurnal cycle, respectively. This allowed gen-

erating the artificial time series using the geostationary-derived time series sampled as it would be acquired by a constellation of polar-orbiting satellites. Further, the artificial CDR was compared to the reference (geostationary-based) dataset to reveal spurious temporal trends in cloud cover introduced by a satellite orbital drift and varying number of satellite observations.

### 2.1   CLARA-A2 CDR derived from NOAA and MetOp polar-oribiting satellites

The Level-2b global daily composite product from the second edition of the CM SAF Cloud, Albedo And Surface Radiation

dataset (CLARA-A2,  Karlsson et al., 2017) was used to derive the AVHRR acquisition times. The CLARA-A2 is derived from the AVHRR global area coverage (GAC) data and consists of observations from the morning satellites: NOAA-12, -15, -17,





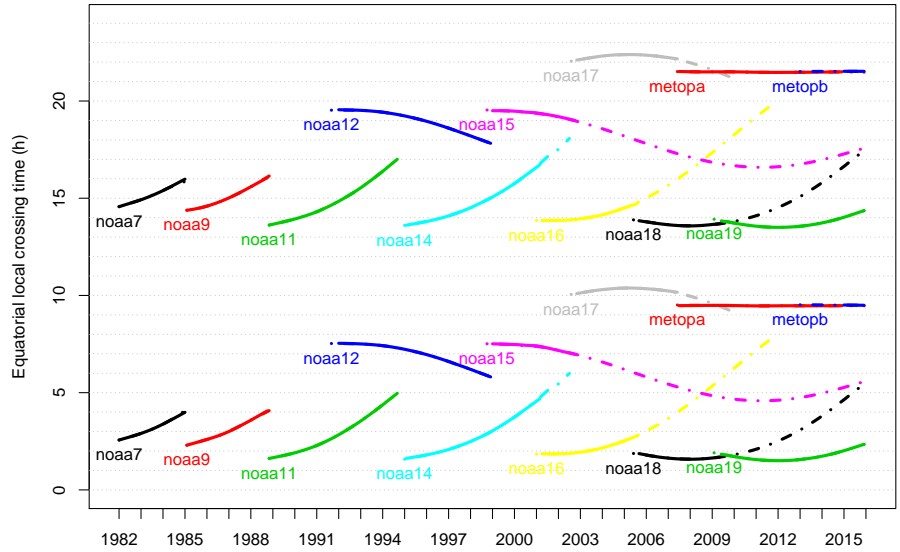

**Figure 1.** Time series of equatorial crossing time of the NOAA and MetOp satellites. Two observations per time per satellite are related to two satellite nodes (ascending, descending). The dot-dashed lines indicate data which are included in the CLARA-A2 dataset but excluded in other CFC CDR (e.g., the Cloud_cci dataset) due to an overlap with other satellites. This exemplifies one source of spurious trends in satellite CDRs caused by different data aggregation strategy.

MetOp-A and MetOp-B, and the afternoon satellites: NOAA-7, -9, -11, -14, -16, -18 and -19. The morning satellites cross the equator on the lit side in the morning as opposed to the afternoon satellites that do so in the afternoon. To derive the Level-2b products, the aggregation methodology proposed by Heidinger et al. (2014) is employed, that for each satellite selects only

two instantaneous AVHRR observations per day (separately for ascending and descending satellite nodes) with the lowest sensor viewing angles. In the ascending node, satellite orbits around the Earth northwards and in the descending node it orbits southwards on the lit side (Ignatov et al., 2004). The Level-2b composites are aggregated from the Level-2a instantaneous retrievals that correspond to a single satellite acquisition. Due to the orbit convergence, the number of acquisitions per satellite per day may vary from 2 at the equator to 14 near the poles.

In spite of some adjustments to the raw AVHRR GAC Level-1b imagery applied during a derivation of the CLARA-A2 Level-2b (e.g. a removal of duplicated and overlapping orbits) (Karlsson et al., 2017), it can be assumed that the selected AVHRR acquisitions times are representative for other CDRs such as Cloud_cci or PATMOS-x. Thus, the results of this study are valid for other AVHRR-based cloud climatologies. Ultimately, the Level-2b CLARA-A2 dataset was used to generate the Level-3 monthly mean cloud fraction composites with and without a distinction between the satellite nodes.

In addition to the AVHRR acquisition times, we used CFC trends observed in the CLARA-A2 CDR for a sake of comparison with the spurious trends estimated in our study.



## 2.2 COMET CDR derived from Meteosat geostationary satellites

The CM SAF Cloud Fractional Cover dataset from Meteosat First and Second Generation (COMET, Stöckli et al., 2017b, 2019) was derived from the MVIRI and SEVIRI imagers aboard a series of Meteosat geostationary satellites. The COMET cloud fraction climatology covers a period 1991–2015 and features the high temporal (30-minute) and low spatial (0.05×0.05 deg) resolutions. It is derived by means of the novel naïve Bayesian classifier (Stöckli et al., 2017a) that features a high accuracy with the overall mean bias below 1% between the COMET CFC and referential SYNOP measurements (Bojanowski et al., 2018). The CFC trends revealed by COMET are consistent with the trends observed in the top-of-atmosphere reflected radiation and surface solar radiation satellite products (Pfeifroth et al., 2018).

Within the study, the cloud fraction diurnal cycles were extracted from the COMET Monthly Mean Diurnal Cycle (MMDC) product. The COMET MMDC has been already validated against the SYNOP cloud observations, and it was proven to be suitable for the analysis of climatic trends and variability in the cloudiness diurnal cycle (Bojanowski and Musiał, 2018). Such accurate dataset was used for the generation of the AVHRR-like synthetic dataset (i.e. out of the COMET dataset), which was further used to quantify the magnitude of the spurious temporal trends.

## 3 Methods

### 3.1 Deriving reference and artificial AVHRR-like CFC time series from the COMET MMDC dataset

To estimate errors and spurious trends in the AVHRR-based CFC CDR induced by the satellite orbital drift and variable number of AVHRR observations a day, the artificial time series was derived from the geostationary COMET CFC dataset sampled at the AVHRR observation times (i.e. COMET CFC "as seen" by the AVHRR sensors). Further, the artificial AVHRR-like CFC was compared with the COMET CFC to estimate errors and spurious trends in the AVHRR-based dataset. The conceptual scheme of the applied methodology is presented in Fig. 2.

To generate AVHRR-like CFC time series, first the COMET CFC MMDC was aggregated to a 0.75×0.75 degree grid by means of the first-order conservative remapping (Jones, 1999; CDO 2018). Then for every grid and month we computed a mean multi-annual CFC diurnal cycle to which we fitted the cubic smoothing spline model (Chambers and Hastie, 1992). The model was further used to predict CFC for the AVHRR overpass times derived from the CLARA-A2 dataset. The retrieved estimates were then averaged to generate the artificial AVHRR-like CFC monthly means. In this step, we created three separate products for: morning satellites (AVHRR.AM), afternoon satellites (AVHRR.PM), and all satellites (AVHRR.AM+PM). Further, the spline model was used to predict CFC at full hours, and these were averaged to generate referential COMET CFC monthly means. The artificial AVHRR-like dataset covering period 1982–2015 and the reference multi-annual CFC datasets were further compared to quantify the errors and spurious trends in the AVHRR-like dataset.

The decision to generate the referential multi-annual CFC by means of the grid-specific spline models instead of using the original COMET CFC MMDC had two premises. Firstly, the COMET time series does not cover years 1982–1990 included in the CLARA-A2 time series, and thus there was a need to substitute these years with the mean climatological diurnal cycles.





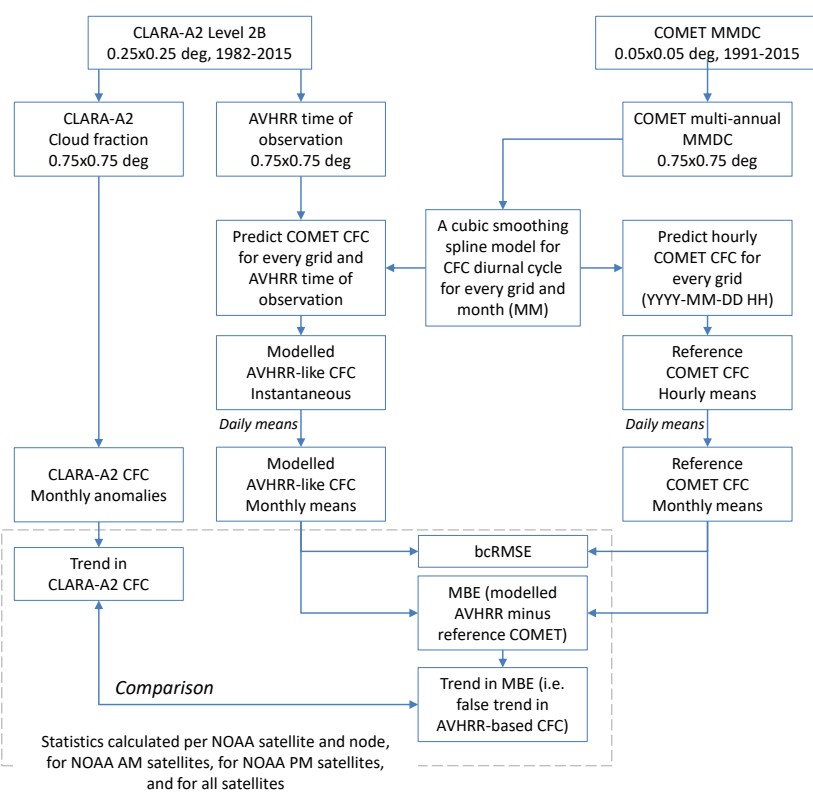

**Figure 2.** Flowchart of the applied methodology consisting of: predicting COMET CFC for AVHRR observation times, deriving COMET-based CFC reference, and calculating errors and magnitude of spurious trends in AVHRR-based CFC caused by the satellite orbital drift.

Secondly, the objective of the study is to analyse spurious trends in the AVHRR CFC CDRs caused by the orbital drift and

sampling issues and not to analyse climatic trends in the CFC diurnal cycles revealed by the COMET dataset. For this please refer to Bojanowski and Musiał (2018).

### 3.2    Assessing errors and spurious trends of the artificial AVHRR-like CFC dataset

To assess the reliability of the artificial AVHRR-like monthly mean CFC dataset, the mean bias error (MBE) as well as the bias-corrected root mean square error (bcRMSE) were computed (see Appendix A for details) between this dataset and the referen-

tial mean monthly COMET CFC. The errors were estimated for each $0.75 \times 0.75$ deg grid, separately for each NOAA/MetOp satellite, with and without the distinction between satellite nodes (ascending, descending). It has to be noted that the estimated error for a single satellite has two components. The first error component is related to how accurately one (for a single node)





or two (for both nodes) discrete AVHRR-based CFC estimates per day represent a daily CFC mean value. Without the orbital drift and climatic change of the CFC diurnal cycle, this error should be stable over the course of a satellite operating time. The

second error component is related to the change of the AVHRR acquisition time induced by the orbital drift and its magnitude varies with the increasing satellite drift. The magnitude of the error depends on the phase and amplitude of the CFC diurnal cycle. If the amplitude is large, the small shift in time of satellite observation will cause an error. Yet, if the amplitude is very small, for instance in a region of constant overcast, even a large change in the observation time does not introduce any error.

To assess the magnitude of spurious temporal trends in the Level-3 AVHRR CFC monthly means, the trend in MBE was

calculated between the reference and artificial AVHRR-like datasets. For this analysis, we used monotonic trends derived using the Theil-Sen estimates (Theil, 1950), and their significance was estimated with the Mann-Kendall test (Kendall, 1938; Mann, 1945). For multiple comparisons of the statistical significance of each grid, we applied the adjustment of the $p$-value using the method of Benjamini and Hochberg (1995). As for the performance assessment, the trends were calculated for individual satellites and nodes, as well as for the three aforementioned synthetic CDRs (AVHRR.AM, AVHRR.PM, AVHRR.AM+PM).

We have excluded from the analysis the MetOp platforms as they do not feature the orbital drift (Fig. 1). Finally, we juxtaposed the theoretical spurious trends in the AVHRR-like Level-3 datasets with the temporal trends derived from the CLARA-A2 CDR.

## 4    Results

### 4.1    Impact of discrete diurnal cycle sampling on the CFC time series

The impact of under-sampling of the cloudiness diurnal cycle on the CFC CDRs is related to representativeness of one (for a single node) or two (for both nodes) observations in respect to the mean daily CFC. The largest positive bias up to 10% is revealed for the nigh-time (2 AM) observations of the afternoon satellites' descending node, whereas the negative bias for afternoon satellites' ascending node (2 PM) and morning satellites' descending node (7 AM) (Fig. 3). The magnitude of bias is similar for all afternoon satellites, because their initial (before-drifting) time of acquisition was similar. Among the morning

satellites, the bias for NOAA-12 and NOAA-15 differs as their initial observation time was 2–3 hours earlier than for the rest of the morning satellites. For the ascending and descending nodes combined, the bias is lower than for the single nodes, which shows that two observations (approx. 12h apart) can substantially better represent the daily CFC than a single observation. Yet, this is partly due to cancelling out the larger negative and positive biases of the individual nodes.

The spatial distribution of the error is similar for all afternoon and morning satellites, and related to CFC diurnal cycle

regimes (Fig. 4). The ascending node of afternoon satellites related to daytime conditions generally reveals a negative bias over the ocean, and a positive one over land (Fig. 5). For the descending node, the spatial pattern is reversed. In both cases, the largest bias (up to ±10%) can be observed over the Southeast and North-east Atlantic. However, in the tropics the bias has the same sign as over the ocean, which is related to a similar phase of the CFC diurnal cycle. For the combined nodes, the bias is largely reduced (up to ±2%) and follows the land-ocean pattern of the afternoon satellites.





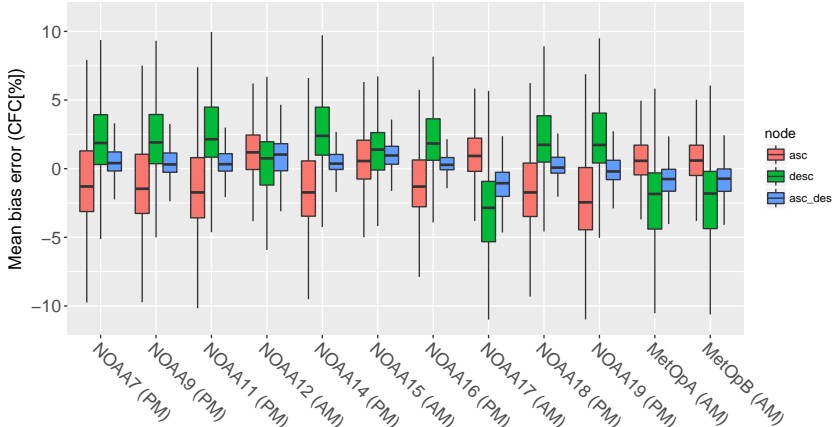

**Figure 3.** Distribution of the mean CFC bias error caused by the discrete sampling of the CFC diurnal cycle presented for each NOAA satellite and each node (asc–ascending, desc–descending, asc_desc–ascending and descending combined). The lower and upper hinges correspond to the 25th and 75th percentiles, while whiskers extend from the hinge to the largest and lowest values within 1.5 times the inter-quartile range.

The 2–3 hour difference in the image acquisition time between NOAA-12 & 15 and the other morning satellites leads to a noticeably different spatial distribution of the error (Fig. 6). The NOAA-12 and NOAA-15 follow the spatial pattern of the afternoon satellites, but with the lower bias values. The NOAA-17 and MetOp platforms show different biases over land (e.g. between Europe and Africa). Moreover, for these satellites a generally greater negative bias for the descending node leads to larger biases for combined ascending and descending nodes.

The bias-corrected root mean square error computed between the AVHRR-like CFC and referential COMET CFC can reach up to 9% due to the under-sampling of the CFC diurnal cycle. The differences between the morning and afternoon satellites are not as evident as for the bias (Fig. 7). For the combined nodes, the average bcRMSE does not exceed 2.5% with the maximum below 4%.

The time of satellite observation does not significantly influence the bcRMSE variability between the sensors. The error for
both morning and afternoon satellites and single satellite node reveals similar spatial distribution with the highest bcRMSE over the Atlantic and over Africa (Fig. 8, 9), where the CFC diurnal cycle has the largest diurnal amplitude (Fig. 4). Nevertheless, these spatial patterns are almost not apparent for the combined satellite nodes (two available observations per day) and the overall bcRMSE is lower. This proves that the CFC CDRs without the distinction between ascending and descending satellite nodes provide significantly more accurate mean monthly estimates.

**4.2   Impact of satellite orbital drift on spurious CFC temporal trends for individual platforms**

The satellite orbital drift induces spurious temporal trends reaching up to ±7% CFC per decade in the AVHRR-like CDRs (Fig. 10). On average, the spurious trends are of 1% per decade both for a single node and the combined nodes. For the afternoon satellites, greater spurious trends are encountered for the ascending than for the descending nodes. The largest



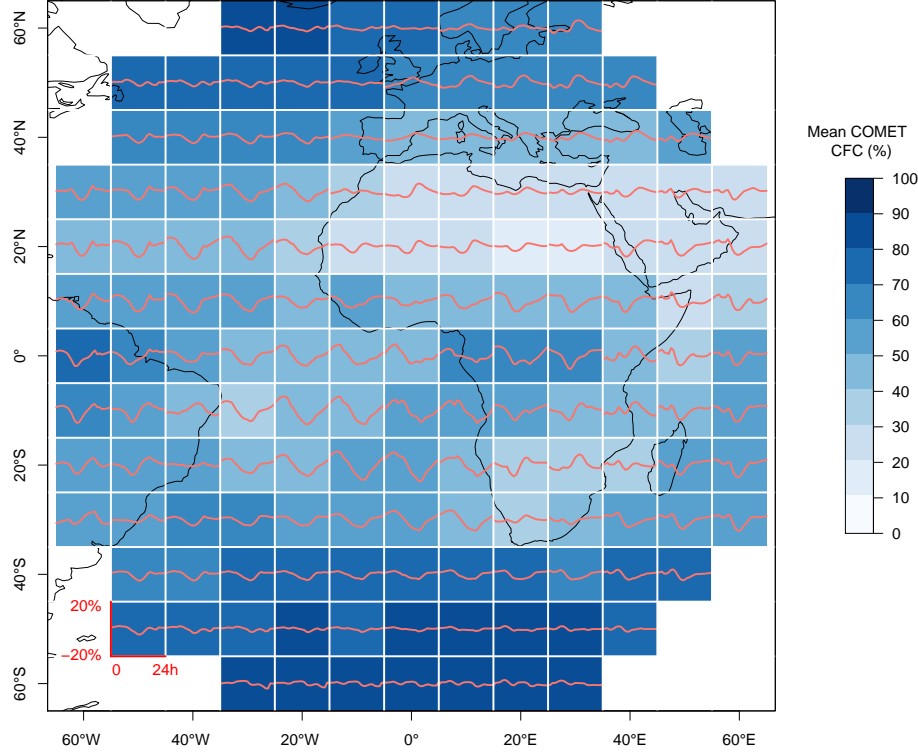

**Figure 4.** CFC diurnal cycle from COMET aggregated in $10 \times 10$ degree grids. Axes used for each grid box are shown in the bottom left corner of the figure: y-axis represents 3-hourly CFC (%) divided by daily mean, x-axis represents local solar time (h). In the blue colour scale the mean COMET CFC is shown.

magnitude of these trends is observed for the NOAA-7 and NOAA-9 satellites due to their quickest orbital drift. Reversely,

CFC time series derived from satellites with the limited orbital drift (e.g. NOAA-19) feature low values of spurious CFC trends. The MetOp platforms were excluded from the analysis as they do not feature the satellite drift.

Figure 11 shows the spatial distribution of the statistically significant spurious temporal trends for the afternoon satellites. It has to be noted that statistical significance of these trends is also affected by the length of the time series of each satellite. For the ascending node the spurious trends are positive over ocean and negative over land. However, for the Southeast Atlantic

the trend is negative, which is related to a different CFC diurnal regime in this area (Fig. 4). The spurious trends for NOAA-14 and NOAA-16 do not reveal this pattern over the ocean. These two satellites drifted more than other afternoon satellites, and their shift in a local time of observation passed a local extreme in the diurnal CFC cycle, which in turn has flattened the temporal trend. The spurious trends for the ascending node dominate the trends observed for the combined nodes, which reveal statistically significant trends with a clear spatial pattern: positive values over ocean (< 6%), and negative over land (> -3%).

For the morning satellites (Fig. 12) the separation of spurious trends between land and ocean is less evident, and absolute values are lower as compared to the afternoon platforms. The positive trends of the descending node dominate the trend





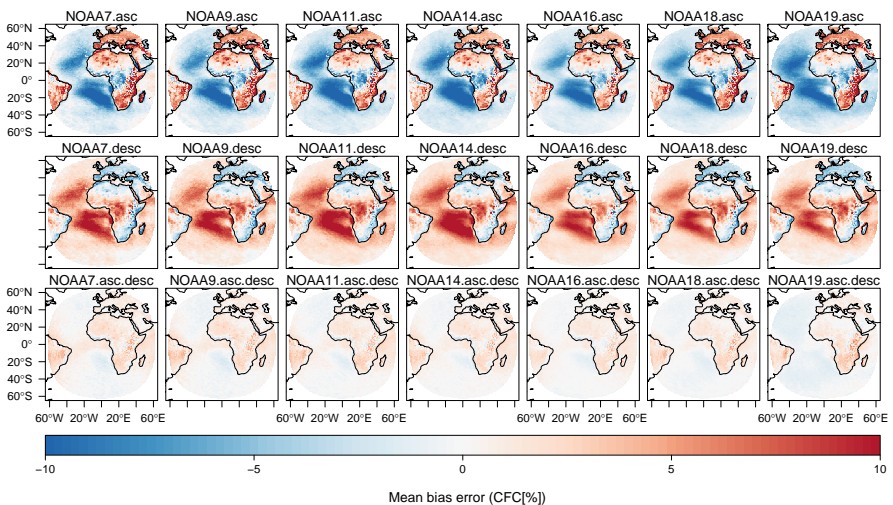

**Figure 5.** Mean bias error of CFC caused by discrete sampling of CFC diurnal cycle presented for each afternoon (PM) NOAA satellite and each node (asc–ascending, desc–descending, asc.desc–ascending and descending combined).

observed for the combined nodes. The NOAA-15 satellite due to its long operational period (almost 20 years) reached the maximal drift, after which it started to return to the initial equatorial local crossing time (Fig. 1). This in turn lowers the overall value of the spurious trend to ±1%.

## 4.3 Impact of satellite orbital drift and variable temporal sampling on CFC CDRs derived from combined AVHRR and MetOps satellite imagery

Cloud cover climate data records derived from a combination of NOAA and MetOp satellites feature: spurious trends of ±1% CFC per decade, up to ±3% MBE and up to 4% bcRMSE, just due to the under-sampling of the CFC diurnal cycle and the orbital drift. These errors are further combined with the cloud retrieval errors. For the CDRs derived from the morning NOAA
satellites, the bias reveals a distinct spatial pattern with positive values over ocean and negative over land (Fig. 13). The opposite spatial pattern is apparent for the afternoon satellites. The CDR derived from the combined morning and afternoon satellites reveals lower MBE and bcRMSE values than the CDR derived from the morning/afternoon satellites separately.

The datasets show similar bcRMSE spatial patterns for the morning and afternoon satellites, however with larger errors in the latter (Fig. 14). The bcRMSE does not exceed 2% in most areas apart from East and South Africa. As for the MBE, the
dataset from combined morning and afternoon satellites reveals the highest performance.

Significant spurious trends up to 1%/dec are observed for the AVHRR-like CDR derived from all morning satellites (Fig. 15) with positive values over Europe, southern Middle East and North-western Atlantic, and negative values over the rest of Atlantic Ocean, Western South America, and Central Africa. This is caused not only by the orbital drift, but by a change in

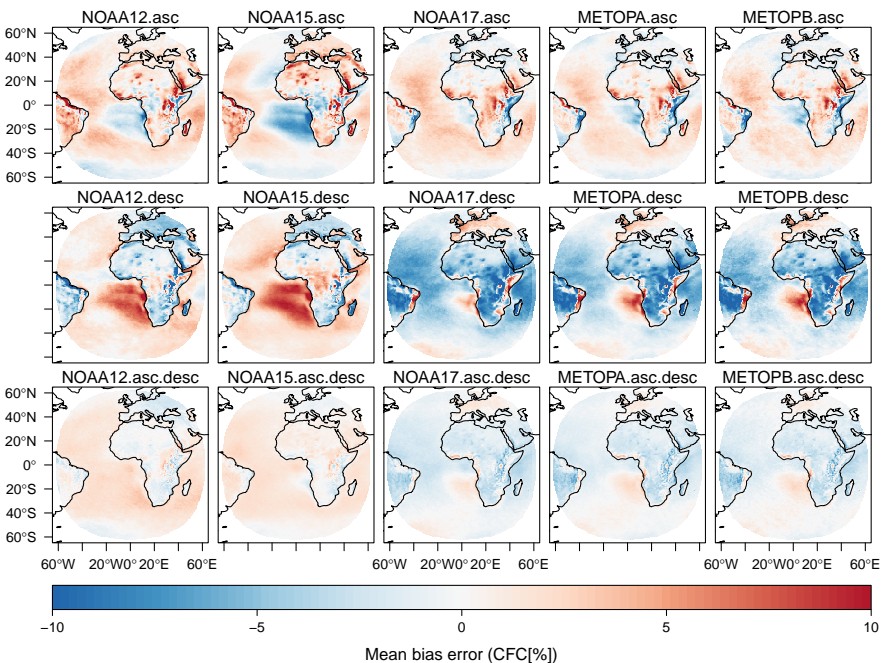

**Figure 6.** Mean bias error of CFC caused by discrete sampling of CFC diurnal cycle presented for each morning (AM) NOAA satellite and each node (asc–ascending, desc–descending, asc.desc–ascending and descending combined).

the observation time between NOAA-12&15 and NOAA-17 and MetOp's. The AVHRR-like CFC CDR from the afternoon
satellites, featuring longer time series than the morning satellites, shows notably lower statistically significant spurious trends below 0.4%/dec. Thus, the CDR derived from combined morning and afternoon satellites is mostly affected by the spurious trends originated from the morning satellites.

Figure 17 depicts the time series of MBE and bcRMSE of AVHRR-like CFC (from all NOAA/MetOp platforms) averaged over the entire Meteosat disc. The overall trend in MBE (i.e. a spurious trend in AVHRR-like CFC) is -0.34%/dec. It is apparent
that this trend is mostly caused by the large inhomogeneity in the MBE time series around 2002. The inhomogeneity renders the CFC derived from AVHRR (starting in 1982) questionable for climate analyses. Yet, as shown in Fig. 15, this is less relevant for some locations. Moreover, after 2002 at least three NOAA satellites have been simultaneously operating. Having two satellite acquisitions per day from each satellite (i.e. from ascending and descending orbits) provides at least 6 observations a day that allow for more correct reconstruction of the cloudiness diurnal cycle.





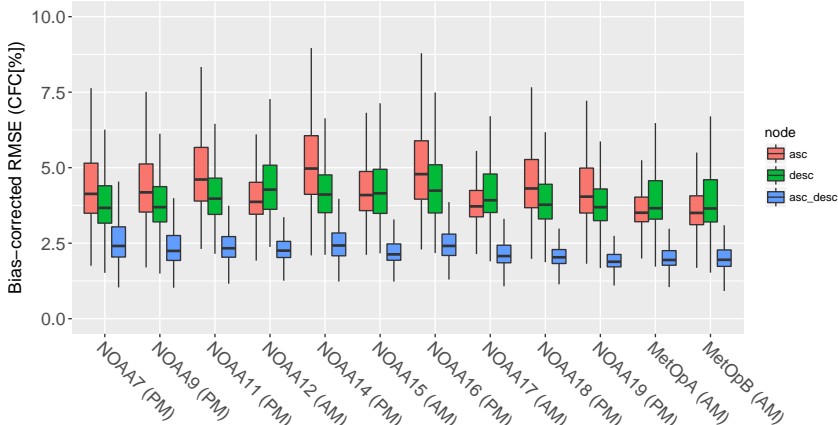

**Figure 7.** Distribution of the CFC bias-corrected root mean square error caused by the discrete sampling of CFC diurnal cycle presented for each NOAA satellite and each node (asc–ascending, desc–descending, asc_desc–ascending and descending combined). The lower and upper hinges correspond to the 25th and 75th percentiles, while whiskers extend from the hinge to the largest and lowest values within 1.5 times the inter-quartile range.

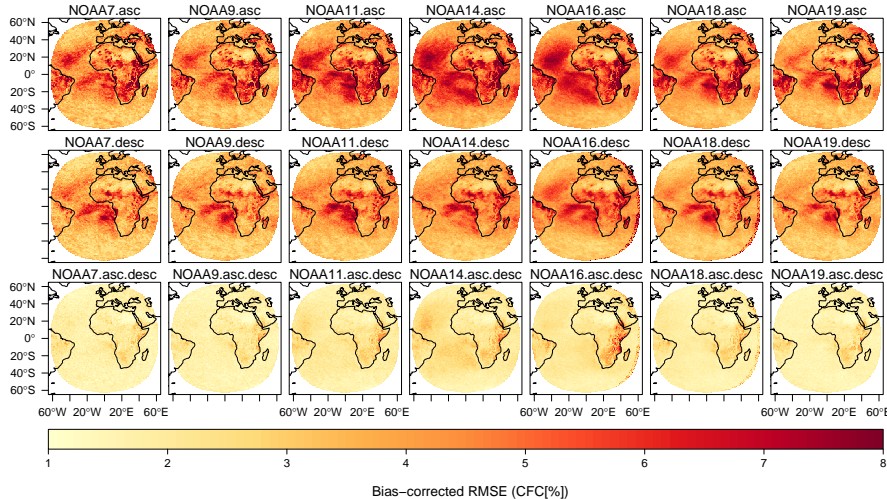

**Figure 8.** Bias-corrected root mean square error of CFC caused by discrete sampling of CFC diurnal cycle, presented for each afternoon (PM) NOAA satellite and each node (asc–ascending, desc–descending, asc.desc–ascending and descending combined).





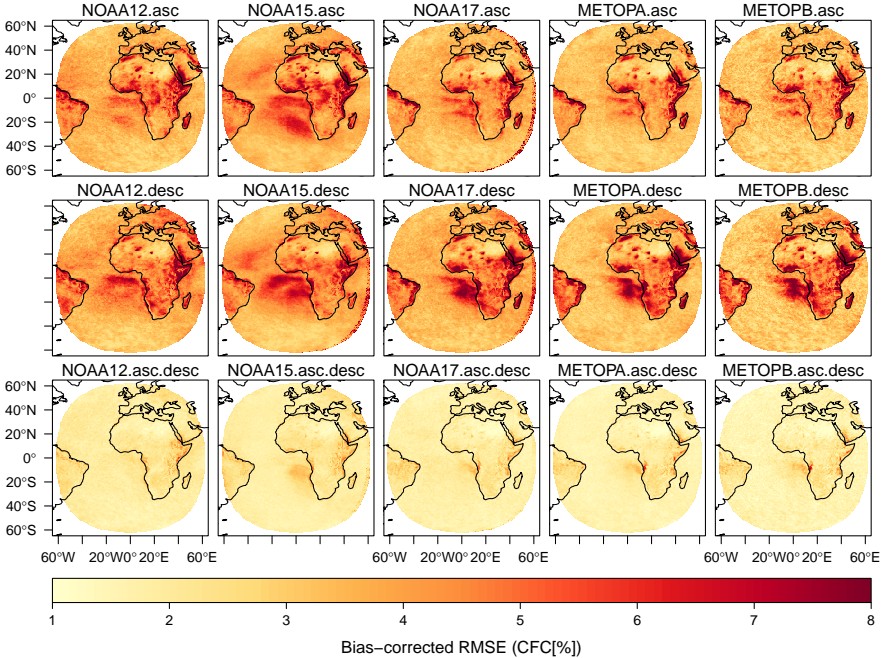

**Figure 9.** Bias-corrected root mean square error of CFC caused by discrete sampling of CFC diurnal cycle, presented for each morning (AM) NOAA satellite and each node (asc–ascending, desc–descending, asc.desc–ascending and descending combined).

## 5 Discussion

This study presents the quantitative assessment of errors and spurious temporal trends in AVHRR-based CDRs induced by under-sampling of the CFC diurnal cycle and NOAA satellite orbital drift. For the individual satellites and specific locations, these CFC errors may reach up to ±10% (MBE), 9% (bcRMSE), and ±7%/dec (spurious trends). The datasets derived from a single satellite are not commonly used in the climate analyses, and usually they feature larger errors than the CDRs derived by combining several platforms. In this respect, the absolute CFC errors for the multi-AVHRR CDR can reach up to ±3% (MBE), 4% (bcRMSE) and ±1%/dec (spurious trends). The distinction between satellite platforms discussed in this study allows for assessment of the CFC errors for a limited period within the AVHRR CDR. This in turn provides a valuable information while selecting NOAA/MetOp satellites and time ranges to be included in a CDR. Furthermore, the distinction between the satellite nodes (ascending/descending) allows for performance assessment separately for the night-time and daytime conditions.

The CFC errors discussed here originate solely from the under-sampling of cloud cover diurnal cycle combined with the satellite orbital drift effect, and as such are not related to the accuracy of the cloud discrimination (masking) on the AVHRR imagery. To assess the accuracy of a cloud mask, the instantaneous satellite observations originating from the Level-2 product are closely collocated with a reference observation to avoid bias caused by the time shift (Bojanowski et al., 2014). Nevertheless, while aggregating the instantaneous measurements to daily or monthly means, the problem of the under-sampling of





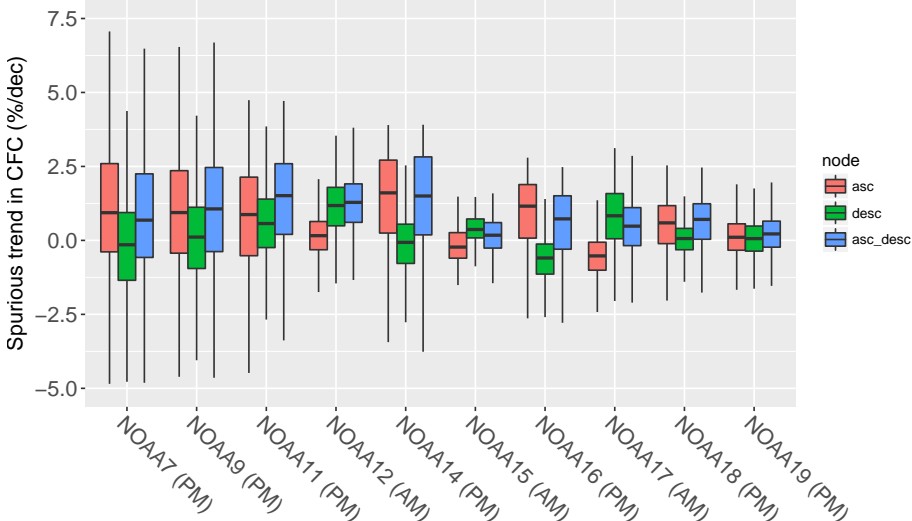

**Figure 10.** Distribution of spurious trends in CFC caused by discrete sampling of CFC diurnal cycle presented for each NOAA satellite and each node (asc–ascending, desc–descending, asc_desc–ascending and descending combined). Trends for MetOp platforms are not presented due to lack of orbital drift. The lower and upper hinges correspond to the 25th and 75th percentiles, while whiskers extend from the hinge to the largest and lowest values within 1.5 times the inter-quartile range.

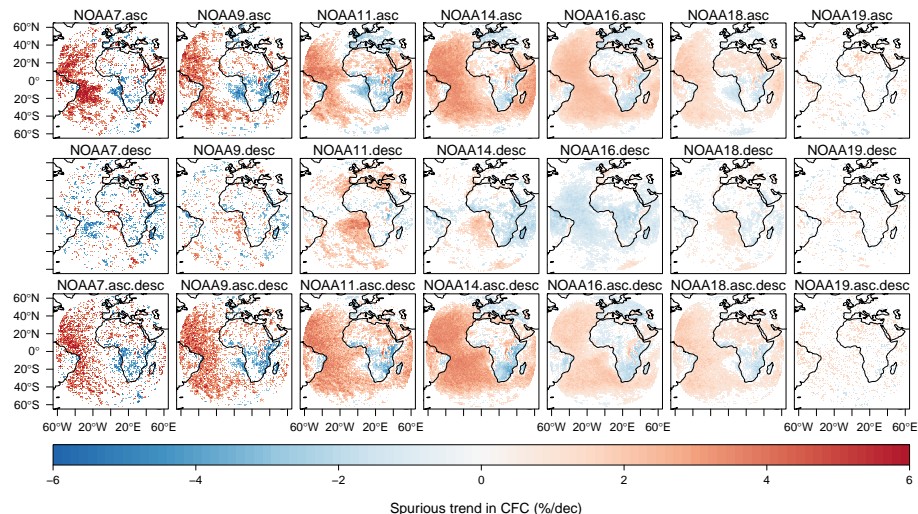

**Figure 11.** Spurious trends in CFC caused by the orbital drift presented for each afternoon (PM) NOAA satellite and each node (asc–ascending, desc–descending, asc.desc–ascending and descending combined). Only statistically significant trends are shown.

a distinct CFC diurnal cycle arises. The number of available observations varies with the number of simultaneously operating satellites. Depending on location- and time-specific diurnal cycle regime, the number of observations may or may not be suf-



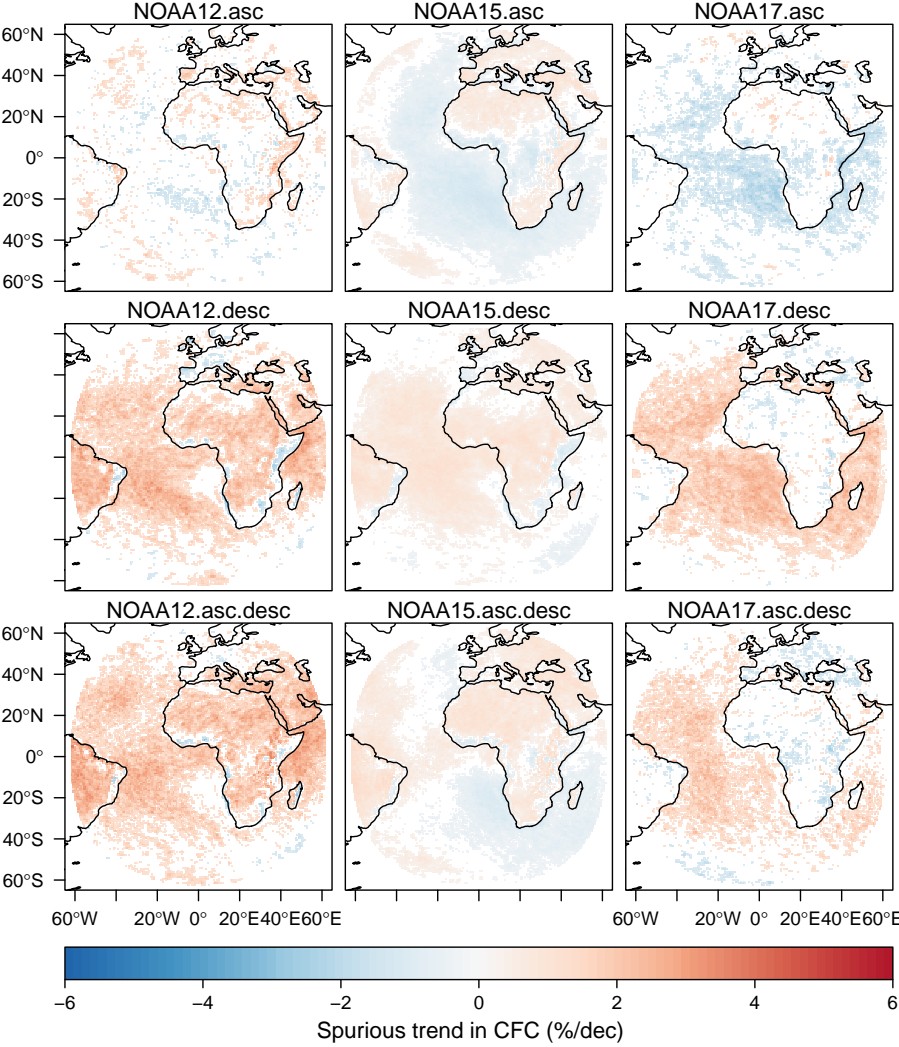

**Figure 12.** Statistically significant spurious trends in CFC caused by the orbital drift presented for each morning (AM) NOAA satellite and each node (asc–ascending, desc–descending, asc.desc–ascending and descending combined). Trends for the MetOp platforms are not presented due to lack of orbital drift.

ficient to represent the CFC daily mean value correctly. This representativeness issue depends on the amplitude of the CFC diurnal cycle. For a small amplitude even a single observation, regardless of the acquisition time, may be enough to represent the mean CFC value. In the case of a large diurnal CFC amplitude, several observations are required to compute the true CFC mean. However, by chance even a single observation that corresponds to the mean estimate might be sufficient in such a case. Ultimately, it has to be emphasised that regardless of the accuracy of the cloud masking algorithm, the under-sampling of the CFC diurnal cycle leads to errors and spurious trends in the aggregated CFC that have been quantified in this study.





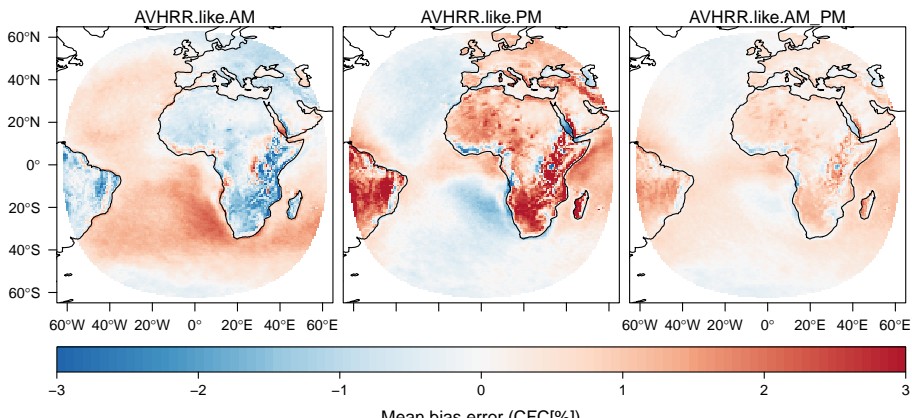

**Figure 13.** The mean bias error caused by orbital drift and discrete sampling of CFC diurnal cycle presented for AVHRR-like CDRs derived from morning (AM), afternoon (PM), and combined (AM_PM) satellites.

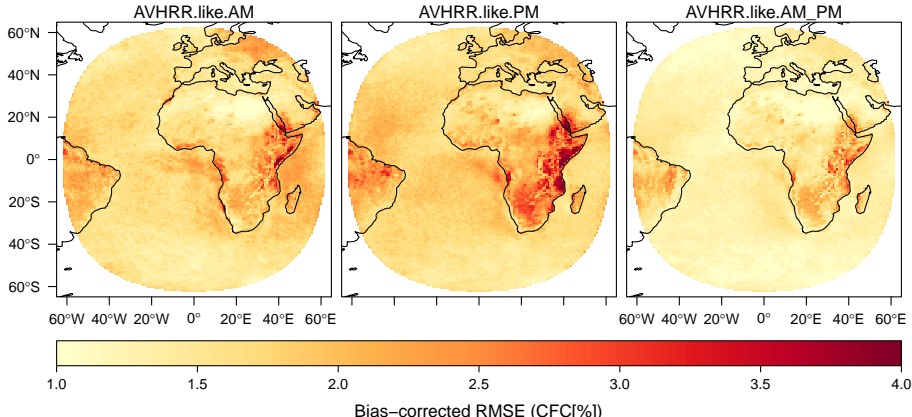

**Figure 14.** Bias-corrected root mean square error caused by orbital drift and discrete sampling of CFC diurnal cycle presented for AVHRR-like CDRs derived from morning (AM), afternoon (PM), and combined (AM_PM) satellites.

The estimated spurious temporal trends in the AVHRR CFC CDR over the Meteosat disc (-0.34% per decade) comply with the GCOS temporal stability requirement of a maximum 1% per decade. Yet, there are regions where the spurious trends exceed

1% per decade and consequently renders the AVHRR-based CFC CDRs not applicable to climate analyses. The GCOS stability requirement is fulfilled by the AVHRR CDR after 2002 due to the increased number of available observations per day acquired by several simultaneously operating satellites. However, in such a case other CDRs (e.g., Platnick et al., 2015; Stengel et al., 2017) derived from MODIS (operating since 2000) offer a better quality of the cloud discrimination (due to improved spectral resolution) and to date do not feature a satellite orbital drift.

The correspondence of the spurious CFC temporal trends computed using artificial AVHRR-like CDR with the trends originating from the CLARA-A2 CDR is moderate. In Fig. 16 we show the observed trends in CFC calculated from the CLARA-A2





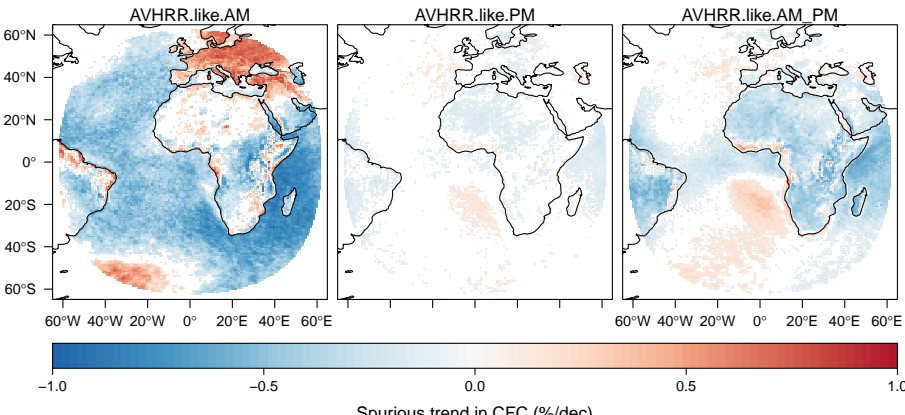

**Figure 15.** Spurious trends caused by orbital drift and discrete sampling of CFC diurnal cycle presented for AVHRR-like CDRs derived from morning (AM), afternoon (PM), and combined (AM_PM) satellites. Only statistically significant trends are shown.

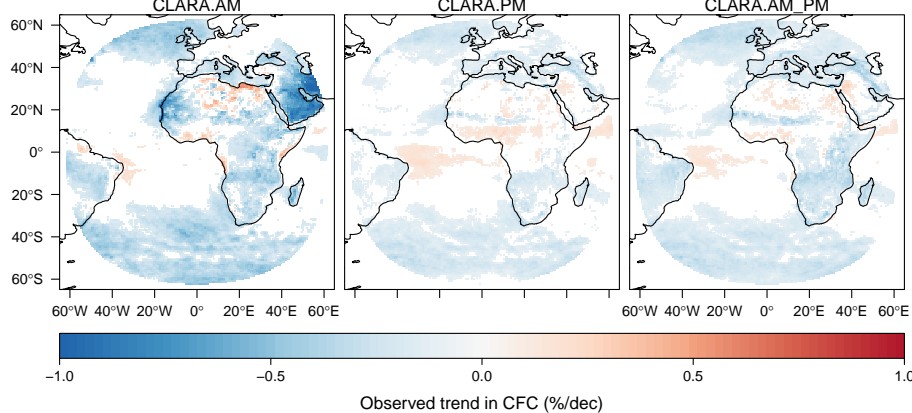

**Figure 16.** Observed trends in the CLARA-A2 CFC CDR derived from morning (AM), afternoon (PM), and combined (AM_PM) satellites . Only statistically significant trends are shown.

L2B for morning and afternoon satellites separately. If regionally similar trends are shown in the observed and spurious trends (Fig. 15), it can be expected that no trend is really occurring there. This is for example a case of the Arabian Peninsula for which a negative CFC trend is observed for morning (AM) satellites, while spurious trends show the same sign and similar

values. Likewise, for a west coast of low-latitudes Africa observations show a positive CFC trend, which is likely not real. On the other hand, in spite of the clear positive spurious trend estimated for Europe, the CLARA-A2 observations do not show any trend. Hence, we expect that a real negative trend in CFC for Europe is hidden by the effect of the under-sampling of CFC diurnal cycle and the satellite orbital drift. This trend would be in line with other findings, for instance of Bojanowski and Musiał (2018) and Pfeifroth et al. (2018). For the afternoon (PM) NOAA satellites, we can expect that observed positive

trend in central Atlantic Ocean is lower than in reality due to the negative spurious trend. Further, the observed positive trend in the tropical Africa may not be real. Although we present the estimated spurious trends as a component of the observed





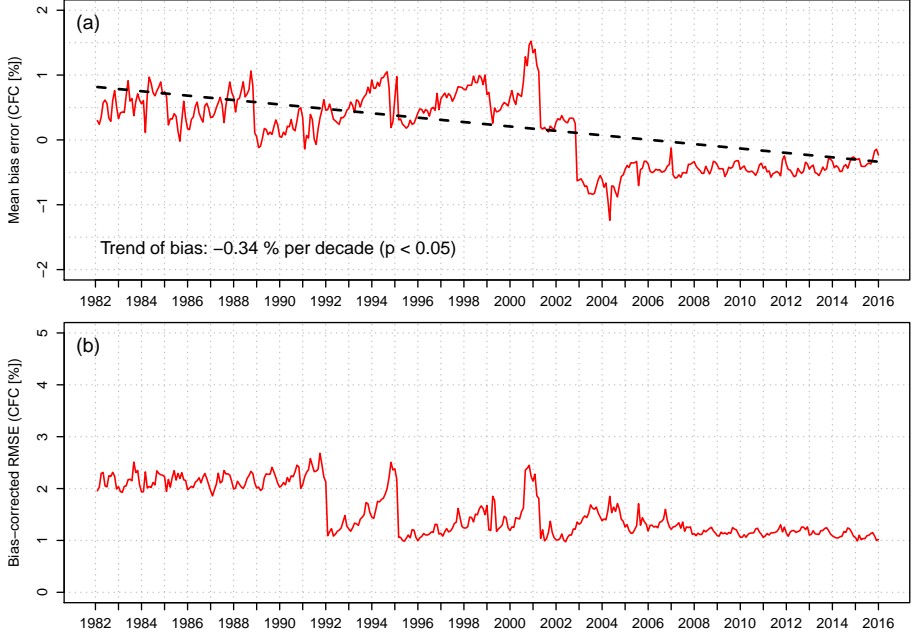

**Figure 17.** Time series of MBE and bcRMSE caused by orbital drift and discrete sampling of the CFC diurnal cycle presented for AVHRR-like CDR derived from combined morning and afternoon satellites.

trends in CDR, we do not recommend a numerical subtraction of spurious from observed trends as a method for the AVHRR-based cloud CDR correction. We refer to the evidence of COMET's good performance and stability, but in our analysis we neglected potential climatological changes in the CFC diurnal cycles, as well as we used averaged monthly mean diurnal cy-

cles. Notwithstanding, we expect that the results presented will allow for a realistic interpretation of the cloud CDRs derived from the AVHRR sensors.

## 6   Conclusions

The Cloud Fractional Cover Climate Data Records (CFC CDRs) generated from the measurements of the AVHRR sensor mounted aboard a series of the NOAA and MetOp polar-orbiting satellites, are subject to errors originating from the under-

sampling of cloudiness diurnal cycle as well from the satellite orbital drift. These errors may lead to spurious temporal trends revealed during climatological analyses. This study provide a unique quantitative assessment of the errors and spurious trends in the AVHRR-based CFC CDRs. For individual NOAA satellites the errors reach up to 10% of MBE and 7% per decade of spurious trends. For the entire data record encompassing all NOAA/MetOp satellites the values are 3% and 1%, respectively. The spurious temporal trend of the AVHRR-like CFC CDR averaged over the Meteosat disc (-0.34 % per decade) complies

with the GCOS temporal stability requirement of a maximum 1% per decade. Yet, there are regions where the spurious trends exceed 1% per decade and consequently renders the AVHRR-based CFC CDRs not applicable to climatic analyses. The GCOS





stability requirement is fulfilled by the AVHRR CDRs after 2002, but it loses the minimum time span of 30 years to be applicable in the climatological analyses, and it can be replaced by the MODIS-based CFC CDRs that feature better spectral resolution and to date do not experience the satellite drift.

The estimated spurious trends were confronted with the CFC trends revealed by the AVHRR-based CLARA-A2 CDR. According to our study, the trends revealed by the CLARA-A2 (and other AVHRR-based CDRs) may be incorrect for several regions. The analysis covered the Meteosat disc due to the spatial extent of the CM SAF COMET dataset used as the reference. The global scale analysis would be beneficial, but it would require application of a climate reanalysis (e.g. ERA-5) which has not been yet proven to resolve the cloud diurnal cycle accurately.

The study was motivated by a need for an improved description of errors incorporated in the CFC CDRs caused by temporal sampling and orbital drift of NOAA/MetOp platforms, which provide one of the longest satellite climatological datasets. We expect that the error estimates will allow for a correct interpretation of AVHRR-based CDRs, reveal a potential improvement of the orbital-drift-corrected dataset, and ultimately will contribute to the development of a yet missing methodology for satellite orbital drift correction that would be commonly applied to the AVHRR-based CDRs. Not only do these include cloud fractional

cover, but other essential climate variables related to cloud physical properties, atmospheric composition, aerosol concentration and surface radiation balance.

**Appendix A: Performance statistics**

Given that $E_k$ is the modelled AVHRR CFC and $M_k$ is the CFC from reference COMET dataset, for the time step $k$, and $n$ is the length of the time series, the performance statistics are defined as:

**Mean bias error:** $\mathrm{MBE} = \frac{1}{n} \sum_{k=1}^{n} (E_k - M_k)$

**Bias-corrected root mean square error:** $\mathrm{bcRMSE} = \sqrt{\frac{1}{n} \sum_{k=1}^{n} (E_k - M_k - \mathrm{MBE})^2}$

*Author contributions.* Conceptualization, J.B.; methodology, J.B. and J.M.; software, J.B. and J.M.; validation, J.B.; writing–original draft preparation, J.B.; writing–review and editing, J.B. and J.M.; visualization, J.B.; project administration, J.B.

*Competing interests.* The authors declare that they have no conflict of interest.

*Acknowledgements.* This research was funded by the National Science Centre Poland under the POLONEZ grant No 2015/19/P/ST10/03990 that received funding from the European Union's Horizon 2020 research and innovation programme under the Marie Sklodowska-Curie grant agreement No 665778. The authors would like to thank the EUMETSAT Satellite Application Facility on Climate Monitoring (https://www.cmsaf.eu/) for providing COMET and CLARA-A2 climate data records.



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
