# Peer review of "Dissecting effects of orbital drift of polar-orbiting satellites on accuracy and trends of cloud fractional cover climate data records"

_Atmospheric Measurement Techniques, 2020_

## Referee Comment (RC1) · Anonymous Referee #1 · 18 Jul 2020

This manuscript addresses the issue that under-sampling and satellite drift makes it difficult to use NPOES satellites in the early record for cloud climate applications. This is an issue that has been pretty well established in previous studies. The main contribution here is the quantification of this error for each satellite and orbit.

The authors mention previous studies that introduced statistical methods of reducing bias due to satellite-drift. I am wondering why the authors did not test these methods in their analysis. Was it out of the scope of this study?

[Figure]

Specific comments:

Line 35: Reference? Line 49: Attitude should be altitude Line 133: Should "full" be changed to "all"?

Lines 136-138: I understand the reasons given for using the spline model instead of the COMET CFC MMDC for the reference data. However, I think some comparison of the spline model to the COMET CFC MMDC should be included to understand the difference choosing this method makes. Also the line: "Firstly, the COMET time series does not cover years 1982–1990 included in the CLARA-A2 time series, and thus there was a need to substitute these years with the mean climatological diurnal cycles." Does this mean the referential dataset was calculated differently before 1990 than after 1990? If this is the case then the impact should probably be discussed.

Line 234: I think Figure 17 (panel A) shows one of the more interesting results of this study. The author's mention inhomogeneity as the reason for the sudden drop, but this is not overly descriptive and a little more discussion would be appreciated. Is the introduction of NOAA-17 (and later the MetOps) at a new sampling time the reason for this sudden shift? Would this figure change substantially if the Metops were left out as they were for other parts of the analysis?

Formatting comment:

The Figures in the manuscript eventually ended up being several pages ahead of where they were first referenced (e.g. Figure 17 is located on page 18 but referenced on page 11). I found myself constantly moving through the document to look at Figures. I wonder if the spacing of the figures could be adjusted?

---

## Referee Comment (RC2) · Anonymous Referee #2 · 23 Aug 2020

In this paper, the authors quantified the uncertainty and magnitude of spurious trends induced by satellite orbit drift in the AVHRR-based cloudness records. The authors estimated that the mean monthly cloud fractional cover of individual NOAA/MetOp satellites reach +/-10%, and the spurious trends reaches +/-7% per decade. For the combined data record, biases of mean and trends is 3% and 1% per decade, respectively. The authors suggest that the AVHRR-derived cloud fraction cover do not comply with the GCOS temporal stability requirement of 1% CFC per decades just due to the orbital drift effect before 2002, while this requirement is fulfilled after 2003. In general,

the paper is well written, and the results are useful for climate studies. The paper might be accepted after addressing the following comments:

(1) In this paper, effect of orbital drift on diurnal cycle has been fully considered. However, the orbital drift could affect cloud cover through other ways, such as changes in solar zenith angle, satellite viewing angle, and orbit altitude. If a same cloud retrieval algorithm is used during the entire satellite operation period, changes in these geometric parameters would result in artificial cloud cover trends. The title of the paper is "Dissecting effects of orbital drift of polar-orbiting satellites", so effect of orbital drift on geometric parameters should also be discussed. (2) The authors might compare the new algorithm in this paper with methods in previous papers, and discuss the advantages and disadvantages of this approach. (3) Line 15: "the time series starting in 2003 is shorter than 30 years that voids climatological analyses." Climatological analyses involve studies of various timescales, so records shorter than 30 years do not void climatological analyses.

---

## Author Comment (AC1) · 10 Sep 2020

This manuscript addresses the issue that under-sampling and satellite drift makes it difficult to use NPOES satellites in the early record for cloud climate applications. This is an issue that has been pretty well established in previous studies. The main contribution here is the quantification of this error for each satellite and orbit. The authors mention previous studies that introduced statistical methods of reducing bias due to satellite-drift. I am wondering why the authors did not test these methods in their analysis. Was it out of the scope of this study?

[Figure]

*The scope of the study was to precisely quantify the effect of the under-sampling and orbital drift on cloud climatology. In our opinion not only does the novelty lie in providing these quantifications for each satellite and orbit, but more importantly in revealing the spatial distribution of errors and spurious trends of the climate data record, and their relation to the GCOS requirements.*

*Extending the study with applications of existing correction methods or proposing a novel one was considered. However, there were two main reasons we decided not to do so:*

*(1) We considered two correction methods to be applied: Devasthale et al. (2012), and Foster and Heidinger (2013) which we explained in lines 60–72. Application of the former is very sensitive to a decision which loadings of the rotated empirical orthogonal function are related to the orbital drift, which introduces a risk of removing the real climatic signal. The latter method assumes that the mean monthly diurnal cycle can be derived as an aggregation of all observations for a given month from the whole record. This is not fully true because there are trends and inter-annual variability in CFC diurnal cycles over the past decades (Bojanowski and Musial, 2018, Yin and Porporato, 2020, doi.org/10.1007/s00382-019-05077-5). Moreover, the correction with this method requires spatial aggregation to a coarse 1 deg resolution to get CFC (in %) from a binary cloud mask. None of the methods was widely accepted and operationally implemented within the major European frameworks providing the satellite-derived geophysical data sets suitable for climate monitoring. We agree that if any new correction method was to be proposed, it would require a benchmarking with the ones above described.*

*(2) Our attempt with developing a novel method that would produce corrected CDR complying with the GCOS requirements was unsuccessful. In our opinion, 2–4 observations per day are not enough to model the CFC diurnal cycle in places where it varies a lot (e.g. in tropics). Of course, the correction methods could involve a fusion with models (e.g. ERA-5 reanalysis) or with CDRs derived from geostationary satellites (e.g. ISCCP, COMET, CLAAS). But then why to correct NPOES-derived CDR,*

*and not to use reanalysis- or geostationary-derived CDRs for the climate applications? One answer could be the higher spatial resolution of NPOES data. Yet, AVHRR-based CDR are based on the 4 km GAC data and anyway are often spatially aggregated during retrieval. The second answer could be the global coverage, i.e. polar regions are not covered by the geostationary data. But, it is also known that AVHRR-based retrievals reveal low performance over the polar regions due to a low number of spectral bands, which is mostly limiting during the polar night. Thanks to the initiatives such as Global Space-based Inter-Calibration System (GSICS), we can expect inter-calibrated GEO-ring-based CDR soon. Moreover, it has to be noted that MODIS-derived climatology, which do not encounter orbital-drift-related errors, cover more than 20 years now. Continued by VIIRS, these two sensors deliver long-term CFC time series (since 1999) which to our believe is longer than the homogeneous part of the AVHRR CDR (since 2003 according to our findings).*

*We hope that, although we do not solve the problem, we can contribute to better understanding of magnitude and spatial distribution of errors and spurious trends in the AVHRR CFC CDRs induced by the satellite orbital drift and the under-sampling of the CFC diurnal cycle. Moreover, our detailed dissection of these errors and trends should be a valuable ancillary information for a correct interpretation of the CDRs.*

Specific comments: Line 35: Reference?

*We added two references.*

Line 49: Attitude should be altitude

*Done.*

Line 133: Should "full" be changed to "all"?

*The spline model was used to predict CFC at hourly temporal resolution, precisely at full hours, i.e. 00:00, 01:00, 02:00, etc. We could rephrase to 'all full hours' but we find it unnecessary (or even misleading).*

Lines 136-138: I understand the reasons given for using the spline model instead of the COMET CFC MMDC for the reference data. However, I think some comparison of the spline model to the COMET CFC MMDC should be included to understand the difference choosing this method makes. Also the line: "Firstly, the COMET time series does not cover years 1982–1990 included in the CLARA-A2 time series, and thus there was a need to substitute these years with the mean climatological diurnal cycles." Does this mean the referential dataset was calculated differently before 1990 than after 1990? If this is the case then the impact should probably be discussed. The referential dataset was calculated in the same way for the whole period. We rephrased this misleading sentence "Firstly, the COMET time series does not cover years 1982–1990 included in the CLARA-A2 time series, and thus there was a need to substitute these years with the mean climatological diurnal cycles." changing 'was a need' to 'would be a need', as this is an explanation of a reason why we did not follow this approach.

*The cubic spline model is fitted strictly to the MMDC not to simplify (smoothen) the cycles. In this context, the 'Reference COMET hourly means' (see the box in Fig. 2) could be directly derived from MMDC. However, if there are any differences between spline model and raw MMDC data, we did not want them to influence the final results. Thus, to be sure this is not the case, we derived both 'Reference COMET hourly means' and 'Modelled AVHRR-like CFC, instantaneous' datasets based on one spline model per grid. The only difference was that in the case of the reference dataset we predicted the values for 'full hours', and for 'AVHRR-like CFC' at exact times of AVHRR overpasses.*

Line 234: I think Figure 17 (panel A) shows one of the more interesting results of this study. The author's mention inhomogeneity as the reason for the sudden drop, but this is not overly descriptive and a little more discussion would be appreciated. Is the introduction of NOAA-17 (and later the MetOps) at a new sampling time the reason for this sudden shift? Would this figure change substantially if the Metops were left out as they were for other parts of the analysis?

*The change in how well the CFC diurnal cycle is described by the CDR occurs when the NOAA-17 is introduced. From 2003 the cycle is described by 6 observations per day, instead of 2 or 4 in preceding years. Fig. 6 and 7 do not reveal that NOAA-17 (with its overpass times) is solely more representative for the mean daily CFC (i.e. the average of two NOAA-17-derived CFCs is closer to the mean daily CFC than from other satellites). Therefore, a reason for this better representation (i.e. lower errors) is more observations a day after 2002 (which we explain in lines 238–239).*

*The Metops were excluded from the per-sensor analysis because they do not drift. We do not see a reason why to exclude them also in Figure 17. Yet, the figure would remain the same – only the lines would end in 2008. We have calculated the trend without years >2008, and it is -0.32% per decade instead of -0.34.*

Formatting comment: The Figures in the manuscript eventually ended up being several pages ahead of where they were first referenced (e.g. Figure 17 is located on page 18 but referenced on page11). I found myself constantly moving through the document to look at Figures. I wonder if the spacing of the figures could be adjusted

*Thank you for this comment, which points out that the paper can be difficult to follow. We do our best to improve it in this respect.*

---

## Author Comment (AC2) · 10 Sep 2020

In this paper, the authors quantified the uncertainty and magnitude of spurious trends induced by satellite orbit drift in the AVHRR-based cloudiness records. The authors estimated that the mean monthly cloud fractional cover of individual NOAA/MetOp satellites reach ±10%, and the spurious trends reaches ±7% per decade. For the combined data record, biases of mean and trends is 3% and 1% per decade, respectively. The authors suggest that the AVHRR-derived cloud fraction cover do not comply with the GCOS temporal stability requirement of 1% CFC per decades just due to the orbital

drift effect before 2002, while this requirement is fulfilled after 2003. In general, the paper is well written, and the results are useful for climate studies.

The paper might be accepted after addressing the following comments:

(1) In this paper, effect of orbital drift on diurnal cycle has been fully considered. However, the orbital drift could affect cloud cover through other ways, such as changes in solar zenith angle, satellite viewing angle, and orbit altitude. If a same cloud retrieval algorithm is used during the entire satellite operation period, changes in these geometric parameters would result in artificial cloud cover trends. The title of the paper is "Dissecting effects of orbital drift of polar-orbiting satellites", so effect of orbital drift on geometric parameters should also be discussed.

*We agree that both satellite orbital drift and sampling at different local times can affect the accuracy of CDRs due to varying sun zenith angle, viewing angle, etc. However, these factors impact the performance of the retrieval, which we intentionally left out of scope of the paper, not to limit to a specific CDR. We used CLARA CDR to derive the NOAA overpass times. Moreover, what we emphasize in the paper is that the quantified errors and spurious trends add up to the errors of the cloud retrieval (line 219). This means that even if the perfect retrieval was developed (i.e. also not sensitive to changing SZA, VZA, etc.), its application to the AVHRR time series would produce a CDR revealing the errors we show in our study.*

(2) The authors might compare the new algorithm in this paper with methods in previous papers, and discuss the ad-vantages and disadvantages of this approach.

*To our knowledge, the concept we used to quantify the errors and spurious trends in AVHRR CDRs was not used before. The main reason for that is probably the novelty introduced by the CM SAF COMET – a long-term, stable and climatologically homogeneous CFC CDR with the resolved diurnal cycle. The method in which the true CFC observations are sampled by the exact AVHRR overpass times is the empirical way to measure the effects of orbital drift and under-sampling on AVHRR-derived CDR. We*

[Figure]

*would prefer to avoid introducing the comparisons with other theoretical methods, because this could distract attention of the reader from the main findings we would like to communicate.*

(3) Line 15: "the time series starting in 2003 is shorter than 30 years that voids climatological analyses." Climatological analyses involve studies of various timescales, so records shorter than 30 years do not void climatological analyses

*We agree that may be an overstatement, because there are surely climatological studies that employ shorter time series. However, we refer here to the GCOS requirements that a 30y+ CDR is required to draw a realistic conclusions on the long-term trends. We rephrased this sentence accordingly.*